# Learning Distances from Data with Normalizing Flows and Score Matching

Peter Sorrenson [1]   Daniel Behrend-Uriarte [1]   Christoph Schnörr [2]   Ullrich Köthe [1]

## Abstract

Density-based distances (DBDs) provide a principled approach to metric learning by defining distances in terms of the underlying data distribution. By employing a Riemannian metric that increases in regions of low probability density, shortest paths naturally follow the data manifold. Fermat distances, a specific type of DBD, have attractive properties, but existing estimators based on nearest neighbor graphs suffer from poor convergence due to inaccurate density estimates. Moreover, graph-based methods scale poorly to high dimensions, as the proposed geodesics are often insufficiently smooth. We address these challenges in two key ways. First, we learn densities using normalizing flows. Second, we refine geodesics through relaxation, guided by a learned score model. Additionally, we introduce a dimension-adapted Fermat distance that scales intuitively to high dimensions and improves numerical stability. Our work paves the way for the practical use of density-based distances, especially in high-dimensional spaces.

## 1. Introduction

Metric learning is fundamental to many machine learning tasks, from clustering to classification. The goal is to determine a distance metric that accurately measures the similarity or dissimilarity between data points. While traditional approaches map data points into spaces where Euclidean distance can be directly applied, these methods are inherently limited by the geometric constraints of Euclidean space, making it impossible to represent arbitrary distance relationships between sets of points.

A more flexible but computationally expensive approach

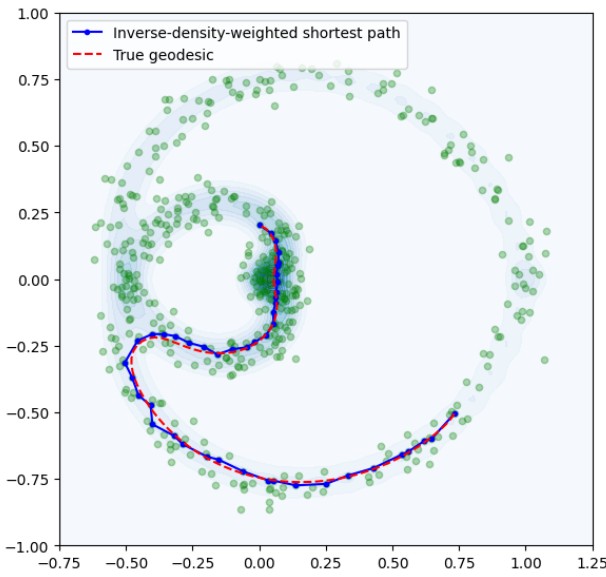

*Figure 1.* Our method learns distance metrics that respect the underlying data distribution. Here, we show shortest paths between two points in a complex Gaussian mixture distribution. The ground truth geodesic (dashed red) and our graph-based approximation (solid blue) both naturally follow high-density regions where data samples (green dots) are concentrated, rather than taking a direct Euclidean path. This behavior is desirable for many machine learning tasks, as it captures the intrinsic structure of the data manifold.

involves defining a Riemannian metric in the data space and solving for geodesic distances. This allows for arbitrary distance relationships, but it requires choosing an appropriate Riemannian metric. Fermat distances (Groisman et al., 2022), which are a type of density-based distance (Bousquet et al., 2003), offer an elegant solution: the metric should be conformal (a multiple of the identity matrix) and inversely proportional to the probability density. Intuitively, this means that distances are stretched in regions where data is sparse and compressed where data is dense. In this way, geodesics pass through high-density regions of the data, respecting any inherent manifold structure (see Figure 1). This approach is consistent with Fermat's principle of least time in optics, where the inverse density functions like a refractive index.

[1]Computer Vision and Learning Lab, Heidelberg University, Germany [2]Image and Pattern Analysis Group, Heidelberg University, Germany. Correspondence to: Peter Sorrenson <peter.sorrenson@gmail.com>.

*Proceedings of the 42nd International Conference on Machine Learning*, Vancouver, Canada. PMLR 267, 2025. Copyright 2025 by the author(s).

This density-based approach to metric learning has evolved significantly over the past two decades. The foundational work of Bousquet et al. (2003) introduced the concept of conformal density-based metrics, though their focus was on semi-supervised learning rather than computing geodesics. Sajama & Orlitsky (2005) took the first steps toward practical implementation by using kernel density estimation and nearest-neighbor graphs for geodesic computation. A breakthrough came from Bijral et al. (2011), who approximated the density as inversely proportional to a power of the Euclidean distance between close neighbors. This elegant approximation sparked numerous theoretical extensions and practical implementations (Chu et al., 2020; Little et al., 2022; Groisman et al., 2022; Moscovich et al., 2017; Alamgir & Von Luxburg, 2012; Mckenzie & Damelin, 2019; Little et al., 2020), becoming known as power-weighted shortest path distances (PWSPD) (Little et al., 2022) or Fermat distances (Groisman et al., 2022; Trillos et al., 2023). While Hwang et al. (2016) provided theoretical convergence guarantees for these methods, notably absent from all previous work is any comparison to ground truth distances, even in simple cases where they can be computed exactly. Our empirical analysis reveals that existing approaches exhibit poor convergence in practice, highlighting a significant gap between theoretical guarantees and practical performance.

Other notable works which use density-based metrics to inform the learning of data manifolds include metric flow matching (Kapusniak et al., 2024) and the Riemannian autoencoder (Diepeveen et al., 2024), which uses a metric defined in terms of score functions.

To establish ground truth geodesics and distances, we develop a numerically stable relaxation method that directly solves the geodesic differential equations. By carefully handling the reparameterization of paths to maintain constant Euclidean velocity and employing appropriate numerical optimization techniques, we can compute accurate geodesics even in challenging cases with large density variations. This enables us, for the first time, to quantitatively evaluate the accuracy of different approximation methods by comparing them to true geodesics in distributions with known densities, such as mixtures of Gaussians.

With this ground truth framework in place, we develop two key improvements to existing methods. First, by employing more accurate edge weights obtained through normalizing flows, we achieve much faster convergence in graph-based approximations. Second, we address the limitations of these graph-based methods in higher dimensions, where paths become increasingly rough due to the curse of dimensionality, by using a relaxation scheme to smooth trajectories. Interestingly, we find that training models via score matching is more effective than using normalizing flows for this purpose, revealing a trade-off between the accuracy of learned log

densities and their gradients (scores).

In summary, our contributions are fourfold: First, we introduce a numerically stable relaxation method to compute ground truth geodesics and distances. Second, we introduce the use of normalizing flows to obtain more accurate edge weights for estimating Fermat distances, achieving significantly faster convergence rates than previous methods. Third, we overcome the limitations of graph-based methods in high-dimensional spaces through a novel smooth relaxation scheme using score matching. Finally, we propose a Fermat distance that naturally scales with dimensionality, making density-based distances practical for real-world, high-dimensional applications. Our approach bridges the gap between theoretical elegance and practical applicability in metric learning.

## 2. Background

### 2.1. Riemannian Geometry

Riemannian geometry provides the mathematical foundation for understanding curved spaces and measuring distances within them. While Euclidean geometry suffices for flat spaces, many real-world datasets lie on curved manifolds where Euclidean distances can be misleading. Riemannian geometry offers powerful tools for analyzing such spaces through the metric tensor, which can encode different geometric structures ranging from positive curvature (like a sphere) to negative curvature (like hyperbolic space), or more complex combinations of both.

More formally, a Riemannian manifold $\mathcal{M}$ is a smooth $n$-dimensional space that locally resembles $\mathbb{R}^n$. In this work, we make the simplifying assumption that $\mathcal{M} = \mathbb{R}^n$. As a result the tangent space $\mathcal{T}_p$, which is a linear approximation of the manifold at $p$, is simply $\mathbb{R}^n$ for all $p$. The metric tensor $g_p : \mathbb{R}^n \times \mathbb{R}^n \to \mathbb{R}$ is a smoothly varying positive-definite bilinear form on the tangent space. It defines an inner product in the tangent space, allowing us to measure lengths and angles.

For a tangent vector $v \in \mathcal{T}_p$, the norm is defined as $\|v\| = \sqrt{g_p(v, v)}$, providing a notion of length in the tangent space. Just as we measure the length of a curve in Euclidean space by integrating its speed, the length of a smooth curve $\gamma(t)$ on the manifold is given by $\int \|\dot{\gamma}(t)\| \, dt$, where $\dot{\gamma}(t)$ is the derivative of $\gamma(t)$ with respect to $t$. This integral accumulates the infinitesimal lengths along the curve, accounting for how the metric varies across the manifold.

The distance between two points on the manifold is defined as the minimal length of a curve connecting the points. The curves that minimize this length are known as geodesics, which are the generalization of straight lines in Euclidean space to curved spaces. These geodesics satisfy a second-

order differential equation known as the geodesic equation (see Appendix A.1). In our context, these geodesics will represent optimal paths through the data space that respect the underlying probability density structure.

## 2.2. Fermat Distances

Having established the framework of Riemannian geometry, we now turn to density-based distances (DBDs), which provide a way to define distances between points sampled from a distribution by utilizing the underlying probability density function. This approach can be particularly useful for tasks such as clustering and pathfinding. If designed well, a DBD can capture desirable properties such as:

1. Shortest paths between points should pass through supported regions of the data. If there is a region with no data between two points, the shortest path should deviate around it to pass through regions with more data.

2. Data should cluster according to its modes. If two data points belong to different modes with little data connecting them, the distance between them should be high.

One way to define a DBD that satisfies these properties is through Fermat distances. Named after Fermat's principle of least time in optics, these distances model the shortest paths between points by considering the inverse probability density (or a monotonic function of it) as analogous to the refractive index.

In the language of Riemannian geometry, we define a metric tensor of the form

$$g_x(u, v) = \frac{\langle u, v \rangle}{f(p(x))^2}, \qquad (1)$$

where $p$ is the probability density, $f$ is a monotonic function, and $\langle \cdot, \cdot \rangle$ is the Euclidean inner product. For example, in a mixture of two Gaussians, this metric would make paths that cross the low-density region between the modes longer than paths that go through high-density regions, effectively capturing the natural clustering in the data. This formulation ensures that regions with higher probability density have lower "refractive index," guiding shortest paths through denser regions of the data. In practice, it is convenient to choose $f(p) = p^\beta$, where $\beta$ is an inverse temperature parameter that controls the influence of the density on the metric. Since our metric enacts a simple scaling of the Euclidean inner product, it is a conformal metric, which locally preserves angles and simplifies key geometric objects such as the geodesic equation (see Lemma A.1 and Theorem A.2).

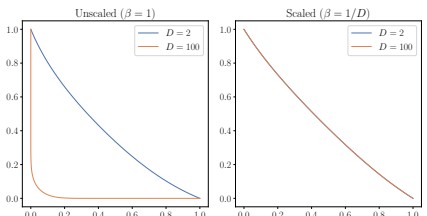

*Figure 2.* Geodesics of the standard normal distribution, projected into 2D, taken between two orthogonal points at distance $\sqrt{D}$ from the origin. The paths are scaled by $\sqrt{D}$ to enable comparison. Left: Unscaled metric (same temperature for all dimensions) leads to sharply curved geodesics in high dimensions. Right: Scaled metric (temperature equal to dimension) leads to consistent geodesics in all dimensions (note that the lines are overlapping).

The length of a smooth curve $\gamma(t)$ is therefore given by

$$L(\gamma) = \int \frac{\|\dot{\gamma}(t)\|}{p(\gamma(t))^\beta} \, dt \qquad (2)$$

This integral accumulates the Euclidean length element $\|\dot{\gamma}(t)\|$, weighted inversely by the density along the path. As a result, segments of the path that pass through low-density regions contribute more to the total length than those through high-density regions.

**Fermat Distances Which Scale with Dimension** While the Fermat distance framework is theoretically elegant, setting $\beta = 1$ (as is common in previous work) leads to unintuitive behavior in higher dimensions. Consider the standard normal distribution in $D$ dimensions. Due to rotational symmetry, the geodesic connecting $x_1$ and $x_2$ will lie in the plane spanned by these two points and the origin. Without loss of generality, suppose this plane aligns with the first two axis directions. The density restricted to this plane is the same as in the two-dimensional case, up to a constant factor. Therefore, geodesics in the $D$-dimensional standard normal distribution are the same as in the two-dimensional standard normal distribution, once we account for the rotation into the plane of $x_1$ and $x_2$.

However, despite this geometric similarity, there's a crucial scaling issue: the typical distance from the origin of points sampled from the $D$-dimensional standard normal distribution is much larger, with a mean of $\sqrt{D}$. As a result, geodesics will start and end in exponentially low-density regions relative to the two-dimensional standard normal. This forces most of the trajectory to make a dramatic detour towards and then away from the high-density region near the origin, creating increasingly extreme curved paths as dimension grows.

To address this issue, if we scale the temperature linearly with dimension ($\beta = 1/D$), the results become consistent with the two-dimensional case (see Figure 2). In Ap-

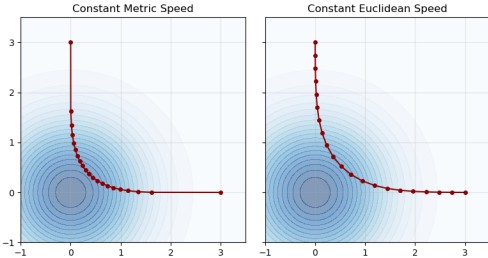

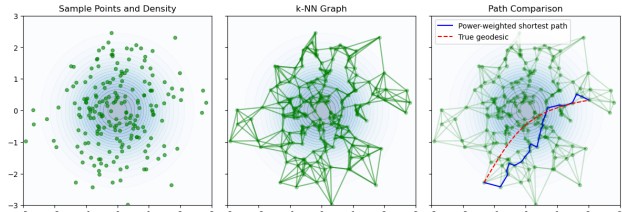

*Figure 3.* Comparison of geodesic parameterizations. Left: Constant metric speed leads to varying Euclidean step sizes, making numerical relaxation challenging. Right: Constant Euclidean speed enables uniform discretization and more stable numerics. Points along the paths indicate discretization steps. Note the slightly different solutions: the left-hand path is approximately 0.1% longer.

*Figure 4.* Illustration of the power-weighted graph approximation. Left: Sample points (dots) and density (gradient) from a 2D standard normal. Center: k-nearest neighbor connections between points, with edge weights determined by cubic Euclidean distances. Right: A shortest path (solid blue) found by Dijkstra's algorithm compared to the true geodesic (dashed red). Note how the discrete path approximates the continuous geodesic by following high-density regions, but is not very smooth.

pendix A.3, we explore this argument in more detail and generalize it to non-Gaussian distributions. This scaling has two key benefits: First, it ensures that geodesic behavior remains intuitive across dimensions, with similar qualitative properties regardless of the ambient dimension. Second, it significantly improves numerical stability by reducing the effect of the probability density, leading to more Euclidean-like geodesics. This effect is clearly visible in Figure 2, where the geodesics are much closer to straight lines in the scaled case.

### 2.3. Geodesic Equation

From the metric, we can compute the geodesic equation, which characterizes the paths of minimum length between points. Just as a straight line in Euclidean space can be described by a second-order differential equation ($\ddot{x} = 0$), geodesics in our curved space satisfy a more complex equation (see Theorem A.2). For a geodesic trajectory $\gamma(t)$, we have:

$$\ddot{\gamma} - 2\beta(s(\gamma) \cdot \dot{\gamma})\dot{\gamma} + \beta s(\gamma)\|\dot{\gamma}\|^2 = 0 \qquad (3)$$

where $s(x) = \frac{\partial \log p(x)}{\partial x}$ is the score of the distribution (the gradient of the log density) and $\|\cdot\|$ is the ordinary Euclidean norm.

The geodesic equation has the property that the speed is constant with respect to the metric, meaning $\sqrt{g(\dot{\gamma}, \dot{\gamma})}$ is constant. While mathematically elegant, this creates numerical challenges: when $\gamma$ passes from a high-probability region to a low-probability one, the Euclidean velocities can differ by orders of magnitude. For example, if the probability density drops by a factor of 1000, the Euclidean velocity must increase by the same factor to maintain constant metric speed. This dramatic speed variation makes standard numerical integrators unstable and requires extremely small step sizes in low-density regions while potentially overshooting in high-density regions.

To address these numerical challenges, we reparameterize the equation to maintain constant Euclidean speed instead of constant metric speed (see Theorem A.3). This simplification is particularly valuable for numerical methods like relaxation, where uniform spacing of points along the trajectory leads to better stability. Figure 3 illustrates how this reparameterization enables uniform discretization compared to the varying step sizes required by constant metric speed. The reparameterized equation differs only slightly from the original (losing just the factor of 2 in the second term):

$$\ddot{\varphi} - \beta(s(\varphi) \cdot \dot{\varphi})\dot{\varphi} + \beta s(\varphi)\|\dot{\varphi}\|^2 = 0 \qquad (4)$$

We stress that, while Equation (3) is easily obtained from standard results on conformal metrics (e.g., Carroll (2004, Appendix G)), the reparameterized form in Equation (4) is, to our knowledge, a novel contribution.

### 2.4. Weighted Graph Algorithm Based on Euclidean Distance

A practical approach to computing Fermat distances is to discretize the problem using sample points. If we have a large number of samples from $p(x)$, we can approximate the density between close neighbors $x_1$ and $x_2$ as constant and hence the shortest path between them as a straight line. Following Bijral et al. (2011), the density is roughly proportional to the inverse of the Euclidean distance, raised to the power $d$, where $d$ is the intrinsic dimension of the distribution:

$$p(\gamma) \propto \frac{1}{\|x_1 - x_2\|^d} \qquad (5)$$

Hence the distance between $x_1$ and $x_2$ according to the metric is approximately proportional to a power of the Euclidean distance

$$\text{dist}(x_1, x_2) \propto \|x_1 - x_2\|^{\beta d + 1} \qquad (6)$$

This elegant relationship suggests a simple algorithmic approach: Bijral et al. (2011) proposes to approximate shortest paths between more distant points by constructing a $k$-nearest neighbors graph of the samples with edge weights given by $\|x_1 - x_2\|^{\beta d+1}$ and applying Dijkstra's algorithm. Figure 4 illustrates the method.

Hwang et al. (2016) gives consistency guarantees for this estimate, proving that a scaled version of it converges to the ground truth distance with a rate of $\exp(-\theta_0 n^{1/(3d+2)})$, with $n$ the sample size, $d$ the intrinsic dimension, and $\theta_0$ a positive constant. Groisman et al. (2022) additionally show that the shortest paths converge to geodesics in a large sample size limit.

While this approach is appealing in its simplicity, can be easily implemented using standard libraries, and comes with theoretical guarantees, the density estimate is too poor to provide accurate estimates of Fermat distances in practice. In the experimental section 4.1.1 we show that paths computed with this approach converge extremely slowly to the ground truth geodesics as sample size increases, even in very simple 2-dimensional distributions such as the standard normal.

This significant gap between theoretical guarantees and practical performance leads us to speculate that the unknown constant $\theta_0$ is very small, resulting in extremely slow convergence despite the exponential form of the bound. This observation motivates our development of more accurate density estimation methods in the following sections. We leave a detailed theoretical investigation of this convergence behavior to future work.

### 2.5. Normalizing Flows

Normalizing flows (Rezende & Mohamed, 2015; Kobyzev et al., 2020; Papamakarios et al., 2021) provide a powerful framework for learning complex probability distributions. The key idea is to transform a simple base distribution (like a standard normal) into the target distribution through a series of invertible transformations. Formally, this is achieved via a parameterized diffeomorphism $f_\theta$ which maps from data to latent space. The probability density is obtained through the change of variables formula:

$$p_\theta(x) = \pi(f_\theta(x)) \left| \det\left(\frac{\partial f_\theta}{\partial x}\right) \right|^{-1} \tag{7}$$

where $\pi$ is a simple latent distribution. A common strategy to design normalizing flows is through coupling blocks (Dinh et al., 2014; 2017; Durkan et al., 2019), where $f_\theta$ is the composition of a series of blocks, each with a triangular Jacobian. This makes the determinant easy to calculate, and hence the above formula is tractable.

Normalizing flows are trained by minimizing the negative log-likelihood on a training set:

$$\mathcal{L} = \mathbb{E}_{p_{\text{data}}(x)}[-\log p_\theta(x)] \tag{8}$$

### 2.6. Score Matching

While normalizing flows provide a way to estimate the density directly, our geodesic equations actually depend on the score function $s(x) = \nabla_x \log p(x)$, which is the gradient of the log-likelihood with respect to spatial inputs. Although one could obtain scores by differentiating the density model, we find that these derived scores are too noisy to be used effectively in practice. We speculate this reflects an inherent trade-off between optimal learning of the log density and its gradient, which we illustrate with simple examples in Appendix C.3. Therefore, we turn to score matching (Hyvärinen, 2005), which offers a direct way to estimate the score function from data, bypassing the need to estimate the density itself.

This family of methods includes several variants such as sliced score matching (Song et al., 2020), denoising score matching (Vincent, 2011) and diffusion models (Song et al., 2021). For our purposes, we use sliced score matching due to its simplicity and ability to estimate the true score, rather than a noisy score as in denoising score matching or diffusion methods.

**Sliced Score Matching** We can learn the score of $p(x)$ by minimizing the score matching objective:

$$\mathbb{E}_{p(x)}[\text{tr}(\nabla_x s_\theta(x)) + \tfrac{1}{2}\|s_\theta(x)\|^2] \tag{9}$$

In high dimensions the trace term is expensive to evaluate exactly, so sliced score matching uses a stochastic approximation (the Hutchinson trace estimator (Hutchinson, 1989)):

$$\mathbb{E}_{p(x)p(v)}[v^T \nabla_x s_\theta(x)v + \tfrac{1}{2}\|s_\theta(x)\|^2] \tag{10}$$

where $v$ is sampled from an appropriate distribution, typically standard normal.

## 3. Methods

### 3.1. Ground Truth Distances through Relaxation

We can solve Equation (4) in several ways to find geodesics between two points. The shooting method starts at one point with an initial velocity guess and integrates the equation forward, iteratively adjusting the initial velocity to reach the target point. Alternatively, the relaxation method discretizes the interval $[0, 1]$ into $n$ equal time steps of size $h = 1/n$, writing $\varphi_i = \varphi(ih)$ for the curve values at these points. Starting from an initial path (either a straight line between endpoints or a more informed guess), it gradually updates the intermediate points while keeping the endpoints

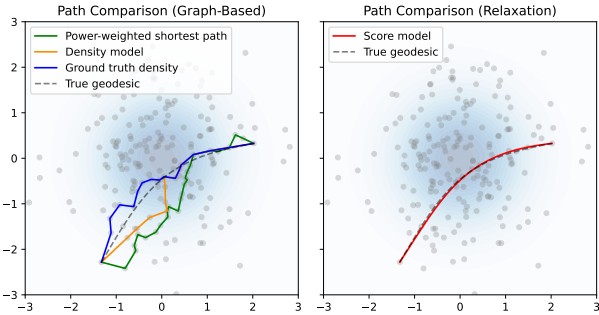

*Figure 5.* Illustration of geodesic approximations by graph-based methods (left) and relaxation methods (right) on 200 points sampled from the 2D standard normal. Left: the power-weighted shortest path method (Section 2.4) produces the roughest path. The shortest path from an inverse-density weighted graph using a model density (Section 3.2) is more direct, but not quite as close to the true geodesic as the equivalent method using the ground truth density. The model is defined as Gaussian with the same mean and covariance as the data. Right: relaxation (Algorithm 1) on the model score function results in a very close approximation to the true geodesic.

fixed to better satisfy the differential equation. Each method has its advantages: shooting is more efficient for exploring geodesics from a starting point, while relaxation is more robust for finding paths between specific endpoints, especially in complex density landscapes where shooting might miss the target. We opt to use relaxation.

Equation (4) leads to the relaxation scheme given in Algorithm 1, which updates each point along the curve based on its neighbors and the local score function. The random ordering of updates helps avoid oscillatory solutions and improves convergence. See Appendix A for a detailed derivation.

### 3.2. Density-Weighted Graph Algorithm

To improve upon previous graph-based methods, we replace their density estimates with those from a normalizing flow trained on the data. For nearby points, we can approximate the geodesic connecting them as a straight line and estimate its length through numerical integration. Specifically, given two points $x_1$ and $x_2$, we estimate their distance as:

$$\text{dist}(x_1, x_2) \approx \sum_{i=1}^{S} \frac{\|y_i - y_{i-1}\|}{p_\theta(y_{i-1/2})^\beta} \tag{11}$$

$$= \frac{\|x_1 - x_2\|}{S} \sum_{i=1}^{S} \frac{1}{p_\theta(y_{i-1/2})^\beta} \tag{12}$$

where $\{y_i\}_{i=0}^{S}$ are equally spaced points along the line from $x_1$ to $x_2$ (i.e., $y_0 = x_1$ and $y_S = x_2$), and $y_{i-1/2}$ denotes the midpoint between consecutive points.

---

**Algorithm 1** One step of relaxation (iterate until convergence)

> **Input:** trajectory $\varphi = \{\varphi_i\}_{i=0}^n$, score function $s$, inverse temperature $\beta$
> $I = \{1, \dots, n-1\}$
> $\texttt{shuffle}(I)$
> **for** $i$ in $I$ **do**
> $\quad v_i = \frac{1}{2}(\varphi_{i+1} - \varphi_{i-1})$
> $\quad w_i = \frac{1}{2}\beta[s(\varphi_i)\|v_i\|^2 - (s(\varphi_i) \cdot v_i)v_i]$
> $\quad \varphi_i = \frac{1}{2}(\varphi_{i+1} + \varphi_{i-1}) + w_i$
> **end for**
> **Return:** $\varphi$

---

**Algorithm 2** Construction of density-weighted graph

> **Input:** data matrix $X$, density function $p$, inverse temperature $\beta$, neighbors $k$, segments $S$
> $\mathcal{G} = \texttt{KNN}(X, k)$ {Construct k-nearest neighbor graph}
> **for** edge $(l, m)$ **in** $\mathcal{G}$ **do**
> $\quad \Delta x = (X_m - X_l)/S$
> $\quad$ **for** $i = 1$ **to** $S$ **do**
> $\quad \quad y_{i-1/2} = X_l + (i - 0.5)\Delta x$
> $\quad$ **end for**
> $\quad \mathcal{G}_{lm} = \Delta x \sum_{i=1}^{S} 1/p(y_{i-1/2})^\beta$
> **end for**
> **Return:** $\mathcal{G}$

---

We use this distance measure to construct a k-nearest neighbors graph, with edge weights given by the approximate geodesic distances rather than Euclidean distances (see Algorithm 2). The resulting sparse graph structure allows us to efficiently compute distances between any pair of points using Dijkstra's shortest path algorithm. Due to the large dynamic range of the density values involved, we perform all distance calculations and graph operations in log space to maintain numerical stability.

### 3.3. Relaxation with a Learned Score Model

The relaxation algorithm described in Section 3.1 requires only the score function $s(x)$, not the density itself. This observation allows us to use a score model trained directly via score matching, bypassing potential issues with density estimation. In practice, we find this approach more effective than computing scores by differentiating a normalizing flow (see experimental results in Section 4).

To ensure robust convergence, we initialize the path using the graph-based method from the previous section. This initialization provides a reasonable approximation of the geodesic, helping the relaxation process avoid local minima and converge more quickly to the optimal path.

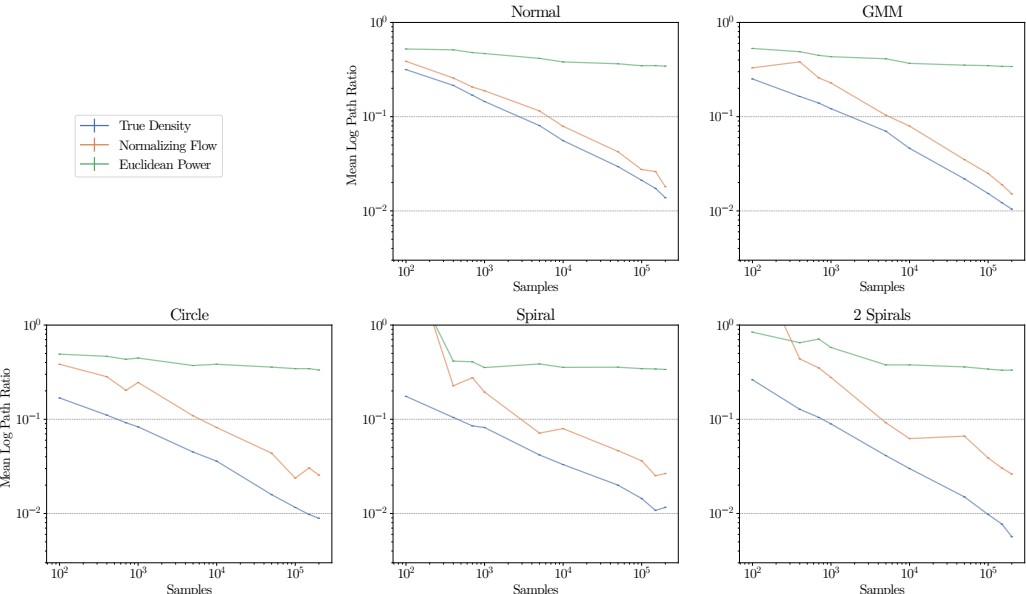

*Figure 6.* The length of paths in learned and ground-truth density weighted graphs (orange and blue) converge to the ground truth at roughly the same rate with increasing sample size, while the power-weighted graph method (green) converges at a far inferior rate.

# 4. Experiments

We have published the code to reproduce these experiments in the following GitHub repository: https://github.com/vislearn/Fermat-Distance.

**Performance Metric**    To evaluate the quality of computed paths, we introduce the Log Path Ratio (LPR) metric that can be applied to any path $\varphi$ connecting two points $x_1$ and $x_2$. We compute the path length $L(\varphi)$ using the ground truth density and compare it to the true geodesic distance:

$$\text{LPR}(\varphi) = \log \frac{L(\varphi)}{\text{dist}(x_1, x_2)} \quad (13)$$

This metric is zero only for the true geodesic and positive otherwise, providing a principled way to evaluate different methods when ground truth densities are available.

## 4.1. Convergence Analysis on 2D Datasets

To evaluate convergence behavior, we analyze five two-dimensional datasets with known ground truth densities, ranging from simple unimodal distributions to complex multimodal mixtures (shown in Figure 6). For each dataset, we compute the average LPR over 1000 randomly sampled paths. Full experimental details and dataset descriptions are provided in the supplementary material.

### 4.1.1. POOR CONVERGENCE OF POWER-WEIGHTED METHODS

We first examine the power-weighted graph method of Bijral et al. (2011), which constructs paths on k-nearest neighbor graphs. Despite theoretical guarantees, we find that performance improves extremely slowly with increasing sample size across all datasets (Figure 6, green lines).

### 4.1.2. DENSITY ESTIMATION IS THE BOTTLENECK

To identify the source of this poor convergence, we developed several variant methods that use different nearest-neighbor density estimators while maintaining the same graph structure (see supplementary material for details). All variants show similarly poor performance, while their counterparts using ground truth densities converge rapidly. This clearly identifies density estimation as the primary bottleneck.

### 4.1.3. IMPROVED CONVERGENCE THROUGH BETTER DENSITY ESTIMATION

Having identified poor density estimation as the core issue, we replace nearest-neighbor estimates with a learned normalizing flow model. Using this density estimate in Algorithm 2, we find that the resulting paths converge at nearly the same rate as those computed using ground truth densities (Figure 6, orange and blue lines). This demonstrates that modern density estimation techniques can effectively close the gap between theoretical guarantees and practical performance. Full experimental details can be found in the

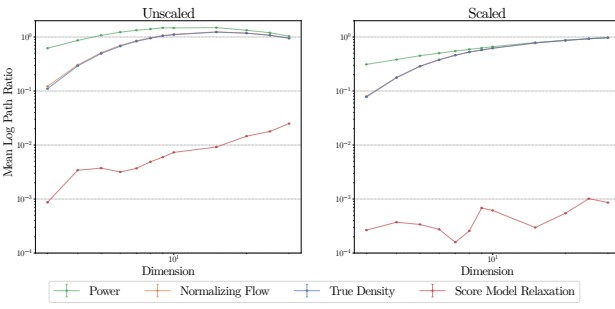

*Figure 7.* Mean LPR with a sample size of 200,000. Graph-based methods (first three) show rapidly degrading performance as dimension increases, even with the scaled metric ($\beta = 1/D$). In contrast, relaxation with a learned score function (red line) maintains good performance across dimensions, effectively avoiding the curse of dimensionality.

supplementary material.

### 4.2. Scaling the Dimension

Machine learning datasets often lie in high-dimensional spaces, making dimensional scaling behavior crucial for practical applications. We investigate this by extending one of our two-dimensional datasets (the standard normal) up to 25 dimensions, testing both the scaled ($\beta = 1/D$) and unscaled ($\beta = 1$) versions of the metric.

#### 4.2.1. GRAPH-BASED METHODS FAIL IN HIGHER DIMENSIONS

All graph-based methods show rapidly degrading performance as dimension increases, regardless of whether we use scaled or unscaled metrics (Figure 7). This failure is a direct consequence of the curse of dimensionality: with fixed sample size but increasing dimension, the probability of finding sample points near the true geodesic becomes vanishingly small. As a result, graph-based paths become increasingly jagged approximations of the smooth ground truth trajectory.

#### 4.2.2. SCORE-BASED RELAXATION OVERCOMES DIMENSIONAL SCALING

To address this fundamental limitation of graph-based methods, we apply our relaxation algorithm (Algorithm 1) with a learned score function. While one might expect to obtain scores by differentiating a trained normalizing flow, we find these derived scores too noisy for reliable convergence (see Appendix C.3 for analysis of this phenomenon). Instead, direct score estimation via sliced score matching provides smoother gradients that enable stable relaxation.

This score-based approach dramatically improves perfor-

mance, maintaining good convergence across dimensions for both scaled and unscaled metrics, though the scaled version ($\beta = 1/D$) shows more reliable behavior (Figure 7). Detailed experimental results and analysis can be found in the supplementary material.

### 4.3. MNIST

We also perform preliminary experiments on MNIST within the latent space of an autoencoder, as described in Appendix C.1. We find that we obtain interpretable and reasonable geodesics connecting digits within this space, and that digits cluster together in an unsupervised way.

## 5. Conclusion

We have presented a systematic study of geodesic computation in density-based metrics, using distributions with known ground truth to rigorously evaluate different approaches. Our investigation reveals several key insights:

First, we demonstrate that previous graph-based methods, despite their theoretical guarantees, converge impractically slowly even for simple distributions like the standard normal. Second, we show that this poor convergence stems primarily from inadequate density estimation rather than the graph approximation itself. Third, we establish that modern deep learning approaches—normalizing flows for density estimation and score matching for gradients—can dramatically improve performance.

However, significant challenges remain. While our methods work well on toy distributions with known densities, extending them to complex real-world data requires further research. Key directions for future work include:

- Unifying normalizing flows and score models into a single framework to improve efficiency

- Developing theoretical understanding of why our methods converge effectively while previous approaches with consistency guarantees do not

- Extending these methods to higher-dimensional densities typical in real applications

This work establishes a foundation for practical density-based distances, bringing theoretical elegance closer to practical applicability in metric learning.

### Impact Statement

This paper presents work whose goal is to advance the field of Machine Learning. There are many potential societal consequences of our work, none which we feel must be specifically highlighted here.

## Acknowledgments

This work is supported by Deutsche Forschungsgemeinschaft (DFG, German Research Foundation) under Germany's Excellence Strategy EXC-2181/1 - 390900948 (the Heidelberg STRUCTURES Cluster of Excellence). The authors acknowledge support by the state of Baden-Württemberg through bwHPC and the German Research Foundation (DFG) through grant INST 35/1597-1 FUGG.

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

# A. Derivations

## A.1. Geodesic Equation

We begin by deriving the geodesic equation for our conformal metric. This is a well-known result in differential geometry (see e.g., Carroll (2004, Appendix G)). The key steps are computing the Christoffel symbols and then simplifying the resulting equation.

**Lemma A.1** (Christoffel Symbols for Conformal Metrics). *For a conformal metric of the form $g_{ij} = \lambda^2 \delta_{ij}$, the Christoffel symbols are:*

$$\Gamma^i_{jk} = \frac{\partial \log \lambda}{\partial x^k} \delta^i_j + \frac{\partial \log \lambda}{\partial x^j} \delta^i_k - \frac{\partial \log \lambda}{\partial x^l} \delta^{il} \delta_{jk} \tag{14}$$

*Proof.* The Christoffel symbols are given by:

$$\Gamma^i_{jk} = \frac{1}{2} g^{il} \left( \frac{\partial g_{lj}}{\partial x^k} + \frac{\partial g_{lk}}{\partial x^j} - \frac{\partial g_{jk}}{\partial x^l} \right) \tag{15}$$

For our metric:

$$\Gamma^i_{jk} = \frac{1}{2\lambda^2} \delta^{il} \left( \frac{\partial \lambda^2}{\partial x^k} \delta_{lj} + \frac{\partial \lambda^2}{\partial x^j} \delta_{lk} - \frac{\partial \lambda^2}{\partial x^l} \delta_{jk} \right) \tag{16}$$

$$= \delta^{il} \left( \frac{\partial \log \lambda}{\partial x^k} \delta_{lj} + \frac{\partial \log \lambda}{\partial x^j} \delta_{lk} - \frac{\partial \log \lambda}{\partial x^l} \delta_{jk} \right) \tag{17}$$

The result follows by simplifying the Kronecker delta products. □

**Theorem A.2** (Geodesic Equation). *For a metric of the form $g_{ij} = p^{-2\beta} \delta_{ij}$, where $p$ is a probability density and $\beta$ is a constant, the geodesic equation in vector notation is:*

$$\ddot{\gamma} - 2\beta(s(\gamma) \cdot \dot{\gamma})\dot{\gamma} + \beta s(\gamma)\|\dot{\gamma}\|^2 = 0 \tag{18}$$

*where $s = \nabla \log p$ is the score function of the probability density.*

*Proof.* For our metric, $\lambda = p^{-\beta}$ and hence $\frac{\partial \log \lambda}{\partial x} = -\beta \frac{\partial \log p}{\partial x} = -\beta s$. Substituting into the formula for Christoffel symbols:

$$\Gamma^i_{jk} = -\beta s_k \delta^i_j - \beta s_j \delta^i_k + \beta s_i \delta^{il} \delta_{jk} \tag{19}$$

The geodesic equation $\ddot{\gamma}^i + \Gamma^i_{jk} \dot{\gamma}^j \dot{\gamma}^k = 0$ then becomes:

$$\ddot{\gamma}^i - \beta s_k \dot{\gamma}^i \dot{\gamma}^k - \beta s_j \dot{\gamma}^j \dot{\gamma}^i + \beta s_l \delta^{il} \delta_{jk} \dot{\gamma}^j \dot{\gamma}^j = 0 \tag{20}$$

which simplifies to the desired vector form. □

**Theorem A.3** (Constant Euclidean Speed Parameterization). *Under a reparameterization that maintains constant Euclidean speed, the geodesic equation becomes:*

$$\ddot{\varphi} - \beta(s(\varphi) \cdot \dot{\varphi})\dot{\varphi} + \beta s(\varphi)\|\dot{\varphi}\|^2 = 0 \tag{21}$$

*Proof.* Let $f : [0,1] \to [0,1]$ be a strictly increasing diffeomorphism and define $\varphi(u) = \gamma(f^{-1}(u))$. The derivatives are related by:

$$\dot{\varphi}\dot{f} = \dot{\gamma} \tag{22}$$

$$\ddot{\varphi}\dot{f}^2 + \dot{\varphi}\ddot{f} = \ddot{\gamma} \tag{23}$$

For constant Euclidean speed, we require:

$$\frac{d}{du}\left( \frac{1}{2}\|\dot{\varphi}\|^2 \right) = \dot{\varphi} \cdot \ddot{\varphi} = 0 \tag{24}$$

Substituting into the geodesic equation and solving for $\ddot{f}$ yields:

$$\ddot{f} = \beta(s(\gamma) \cdot \dot{\varphi})\dot{f}^2 \tag{25}$$

The result follows by substituting back and simplifying. □

## A.2. Relaxation Scheme

The central finite difference approximations for the first and second derivatives are

$$\dot{\varphi}_i \approx \frac{\varphi_{i+1} - \varphi_{i-1}}{2h} \tag{26}$$

and

$$\ddot{\varphi}_i \approx \frac{\varphi_{i+1} - 2\varphi_i + \varphi_{i-1}}{h^2} \tag{27}$$

By substituting this into the differential equation (Equation (21)) we have

$$\frac{\varphi_{i+1} - 2\varphi_i + \varphi_{i-1}}{h^2} - \beta(s(\varphi_i) \cdot \dot{\varphi}_i)\dot{\varphi}_i + \beta s(\varphi_i)\|\dot{\varphi}_i\|^2 \approx 0 \tag{28}$$

where $\dot{\varphi}_i$ is estimated using finite differences. We can rearrange this to

$$\varphi_i \approx \frac{\varphi_{i+1} + \varphi_{i-1}}{2} + \frac{h^2\beta}{2}\left(s(\varphi_i)\|\dot{\varphi}_i\|^2 - (s(\varphi_i) \cdot \dot{\varphi}_i)\dot{\varphi}_i\right) \tag{29}$$

and use the equation as an update rule, updating each position of the curve except the endpoints at each iteration.

Dividing by very small $h$ could lead to numerical instability. We can avoid dividing and multiplying by $h$ by the following update rule. First define

$$v_i = \frac{\varphi_{i+1} - \varphi_{i-1}}{2} \tag{30}$$

then update with

$$\varphi_i = \frac{\varphi_{i+1} + \varphi_{i-1}}{2} + \frac{\beta}{2}\left(s(\varphi_i)\|v_i\|^2 - (s(\varphi_i) \cdot v_i)v_i\right) \tag{31}$$

## A.3. Justification of Scaling Fermat Distances with Dimension

We propose to scale the metric using a temperature equal to the dimension of the data. In order to justify this, consider the behavior of a generalized Gaussian distribution of the form $p(x) \propto \exp(-\|x\|^\alpha/\alpha)$. It can be shown that $\mathbb{E}[\|x\|^\alpha] = D$ and thus the typical distance from the origin of points sampled from $p$ scales as $D^{1/\alpha}$. The score in this case is $s(x) = -\|x\|^{\alpha-2}x$.

Due to the rotational symmetry of the distribution, we can suppose without loss of generality that a sample $x$ lies in the plane of the first two coordinates and hence the problem reduces to the 2-dimensional one. However, since the typical distance of sampled points scales with the dimension, in order to avoid very exaggerated geodesics in high dimensions which deviate sharply towards the origin, we need to scale the density appropriately. By rescaling $\gamma$ in Equation (3) by $D^{1/\alpha}$, the score scales like $D^{(\alpha-1)/\alpha}$, and so the terms in the equation scale as:

$$\ddot{\gamma} - 2\beta(s \cdot \dot{\gamma})\dot{\gamma} + \beta s\|\dot{\gamma}\|^2 \to D^{1/\alpha}\ddot{\gamma} - 2\beta D^{(\alpha+1)/\alpha}(s \cdot \dot{\gamma})\dot{\gamma} + \beta D^{(\alpha+1)/\alpha}s\|\dot{\gamma}\|^2 \tag{32}$$

$$= D^{1/\alpha}\left(\ddot{\gamma} - 2\beta D(s \cdot \dot{\gamma})\dot{\gamma} + \beta D s\|\dot{\gamma}\|^2\right) \tag{33}$$

Therefore the geodesic equation can scale consistently if, and only if, $\beta = 1/D$. The same reasoning applies to the reparameterized Equation (4).

We can apply a similar argument to the Student-t distribution of the form $p(x) \propto (1 + \|x\|^2/\nu)^{-(\nu+D)/2}$ with $\nu > 2$. In the limit $\nu \to \infty$, the distribution becomes normal, and otherwise is characterized by heavier tails. The expected squared norm $\mathbb{E}[\|x\|^2] = \nu D/(\nu - 2)$, so the typical distance from the origin scales like $\sqrt{D}$, as in a Gaussian distribution. The score function has the form $-\frac{\nu+D}{\nu+\|x\|^2}x$, and with $x \sim \sqrt{D}$, we have $s \sim x \sim \sqrt{D}$, again the same behavior as the Gaussian. Hence the scaling behavior is approximately the same as the Gaussian distribution, so the above argument for $\beta = 1/D$ applies as well. Given that the same scaling behavior holds as in the generalized Gaussian, it suggests that $\beta = 1/D$ is a natural default choice across a wide range of distributions.

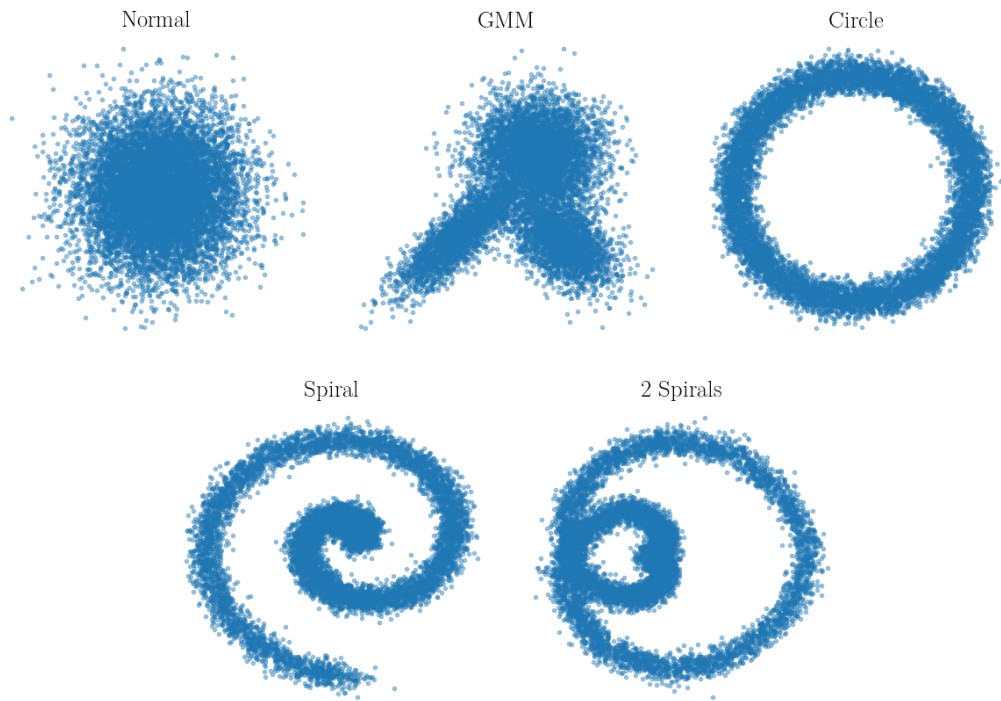

*Figure 8.* Samples from the 5 two-dimensional datasets.

# B. Experimental Details

## B.1. Software Libraries

We used the `numpy` (Harris et al., 2020) and `scipy` (Virtanen et al., 2020) Python libraries for basic numerical and graph operations and `matplotlib` (Hunter, 2007) for plotting. We used `scikit-learn` (Pedregosa et al., 2011) to compute nearest neighbor graphs and `pytorch` (Paszke et al., 2019) for training neural networks. We used `FrEIA` (Ardizzone et al., 2018-2022) for constructing normalizing flows.

## B.2. Two-Dimensional Datasets

We use the following two-dimensional datasets. All are implemented as Gaussian mixture models. The first two are explicitly Gaussian, the final three are GMMs with 50 components fitted using `scikit-learn` to data generated by adding noise to certain geometric structures.

1. Standard normal distribution

2. Gaussian mixture with 3 components:
   - Means: $\mu_1 = (2, 1.4)$, $\mu_2 = (6.5, 6.3)$, $\mu_3 = (8, 1)$
   - Covariances: $\Sigma_1 = \begin{pmatrix} 3 & 2.5 \\ 2.5 & 3 \end{pmatrix}$, $\Sigma_2 = \begin{pmatrix} 3 & 0 \\ 0 & 3 \end{pmatrix}$, $\Sigma_3 = \begin{pmatrix} 2 & -0.8 \\ -0.8 & 2 \end{pmatrix}$
   - Weights: $(0.25, 0.5, 0.25)$

3. Circle with radius 1 and Gaussian noise ($\sigma = 0.08$)

4. Single spiral with 1.75 rotations ($3.5\pi$ radians) and Gaussian noise ($\sigma = 0.05$)

5. Two interleaved spirals with 1 rotation each and Gaussian noise ($\sigma = 0.045$)

The spirals reach a radius of 1 after a single rotation.

### B.3. Normalizing Flow Training

We used the `FrEIA` library (Ardizzone et al., 2018-2022) to design normalizing flows using spline coupling blocks (Durkan et al., 2019). We used the following hyperparameters:

- Blocks: 12 for two-dimensional data, 5 for standard normal in $D$ dimensions

- Spline bins: 10

- Domain clamping in the splines: 5

- ActNorm between blocks

- Noise of $10^{-5}$ added to the training data

- LR = $5 \times 10^{-4}$

- Weight decay = $10^{-6}$

- Fully-connected coupling block subnets with 3 hidden layers of 64 dimensions each, with ReLU activation and BatchNorm

- Training iterations (as a function of training set size $n$): $2500 \times 2^{n//1000}$

- Batch size = 256 if $n > 400$, else $\lfloor n/3 \rfloor$

- Adam optimizer

### B.4. Score Model Training

We use fully connected networks to predict the score, with 6 hidden layers of varying width depending on the input dimension. The architecture consists of:

- Input layer $\rightarrow$ 6 hidden layers $\rightarrow$ Output layer (dimension matching input)

- Hidden layer widths:
    - $3 \leq D \leq 5$: 128
    - $6 \leq D \leq 8$: 170
    - $9 \leq D \leq 25$: 200

- LR = $10^{-3}$

- Weight decay = $10^{-6}$

- Softplus activation function

- Training iterations (as a function of training set size $n$): $2500 \times 2^{n//1000}$

- Batch size = 256 if $n > 400$, else $\lfloor n/3 \rfloor$

- Adam optimizer

In all cases, the chosen model checkpoint was that which achieved the lowest training loss.

### B.5. Compute Resources

For the two-dimensional datasets, we performed all experiments on a machine with a NVIDIA GeForce RTX 2070 GPU and 32 GB of RAM. It took approximately 24 hours to run all experiments.

For the higher dimensional standard normal distributions, we performed all experiments on a machine with two GPUs: a NVIDIA GeForce RTX 2070 and a NVIDIA GeForce RTX 2080 Ti Rev. A. The machine has 32 GB of RAM. It took approximately 24 hours to run all experiments.

In addition we did some preliminary experiments on the first machine, totalling approximately 48 hours of compute time.

# C. Additional Experimental Results

## C.1. MNIST experiments

*Table 1.* For each digit, we compute the mean and standard deviation of the log distances between the cluster mean and 20 randomly sampled points of that class (intra-class log distance), as well as between the cluster mean and 20 randomly sampled points of other classes (inter-class log distance). We also compute the difference, which gives an indication of how well-separated the digits are in terms of log distance.

| Digit | Intra-Class Log Distance | Inter-Class Log Distance | Difference (Inter–Intra) |
|---|---|---|---|
| 0 | 2.41 ± 0.40 | 3.73 ± 0.27 | 1.32 ± 0.49 |
| 1 | 1.69 ± 0.49 | 3.73 ± 0.31 | 2.04 ± 0.58 |
| 2 | 3.56 ± 0.29 | 4.05 ± 0.19 | 0.49 ± 0.35 |
| 3 | 3.03 ± 0.33 | 3.78 ± 0.24 | 0.75 ± 0.41 |
| 4 | 3.04 ± 0.38 | 3.77 ± 0.36 | 0.73 ± 0.53 |
| 5 | 3.43 ± 0.35 | 4.02 ± 0.18 | 0.59 ± 0.39 |
| 6 | 2.89 ± 0.41 | 3.79 ± 0.31 | 0.91 ± 0.51 |
| 7 | 2.86 ± 0.47 | 3.81 ± 0.30 | 0.95 ± 0.56 |
| 8 | 3.29 ± 0.33 | 3.74 ± 0.21 | 0.46 ± 0.39 |
| 9 | 2.76 ± 0.45 | 3.56 ± 0.38 | 0.80 ± 0.59 |

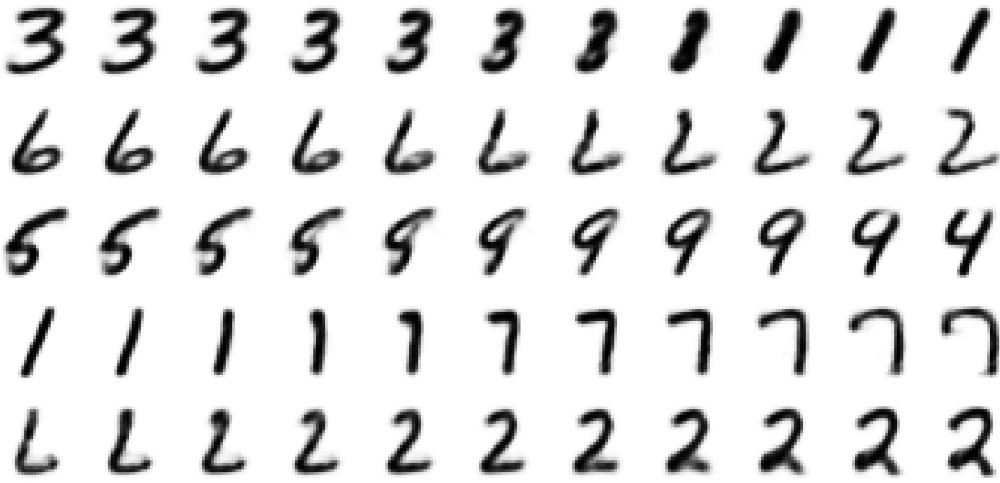

*Figure 9.* Each row visualizes an MNIST geodesic, obtained by relaxation (Algorithm 1) using a learned score model within a 10-dimensional autoencoder latent space.

We train an autoencoder to compress MNIST (LeCun et al., 2002) digits to a 10-dimensional space and subsequently train a sliced score matching model (Song et al., 2020) and normalizing flow (Dinh et al., 2017) on the latent distribution. Both encoder and decoder use fully connected layers with ReLU activations, with a single hidden layer of width 400. The score model has input and output dimension 10 and has hidden layers of width 128, 256, and 128, with SiLU activations. The normalizing flow is built with the FrEIA framework (Ardizzone et al., 2018-2022), made up of 4 affine coupling blocks, with a permutation between each block. Each block uses a coupling network with 2 hidden layers of width 128 and SiLU activations. All models are trained for 400 epochs with the Adam optimizer, the autoencoder with learning rate $10^{-3}$ and the score model and normalizing flow with learning rate $10^{-4}$.

To compute distances between two points sampled from the latent distribution, we the same procedure as in the main text: i) construct a density-weighted graph (Algorithm 2) from the normalizing flow model ii) use Dijkstra's algorithm to find the shortest path between the points according to the graph iii) solve the relaxation problem (Algorithm 1) with the score model, using the graph-based path as initialization iv) evaluate the length of the relaxed path using the normalizing flow model,

according to Equation (2).

In Table 1 we report mean and standard deviation of log distances between each digit cluster mean and 20 random samples of the same or other digits. We see that same-class distances are always lower than other-class distances, up to a safe margin of error. We also make the following observations: 1s cluster tightly, likely due to having fewer active pixels and therefore having higher likelihoods; 8s are more ambiguous, likely due to easy deformations into other digits (it is quite easy to remove some pixels to turn an 8 into a 3, 5, 6 or 9). We leave deeper analysis of Fermat distances on MNIST to future work.

In Figure 9 we visualize geodesics between 5 randomly sampled pairs from the latent distribution. The geodesics are solved in the latent space and the resulting latent points are decoded to image space for visualization. We see that the geodesics follow high-density regions of the distribution, leading to interpretable and reasonable transitions between the digits.

## C.2. Convergence of Alternative Graph Edge Weightings

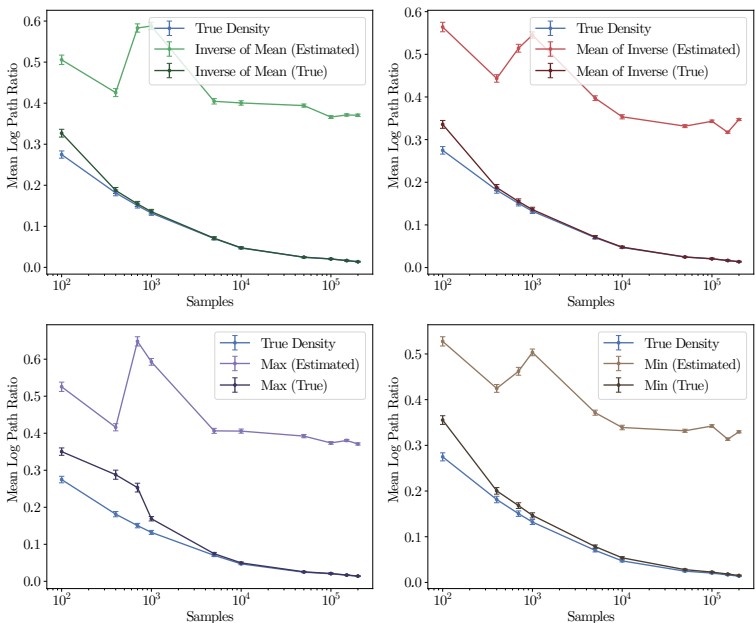

*Figure 10.* Alternative edge weightings based on the nearest neighbor density estimator converge poorly. The same weightings based on the ground truth density converge well. Experiments performed on the GMM two-dimensional dataset.

We try out other edge weights, as described in the main text. Given the nearest-neighbor density estimates $\hat{p}(X_l) \propto (\min_m \|X_l - X_m\|)^{-d}$ for data points $X_l$, we approximate the graph edges by $\mathcal{G}_{lm} = \|X_l - X_m\|/\tilde{p}((X_l + X_m)/2)$ where $\tilde{p}$ is a density estimate for the midpoint. We use 4 variants:

1. **Inverse of Mean:** $\tilde{p}((X_l + X_m)/2) = (\hat{p}(X_l) + \hat{p}(X_m))/2$

2. **Mean of Inverse:** $1/\tilde{p}((X_l + X_m)/2) = (1/\hat{p}(X_l) + 1/\hat{p}(X_m))/2$

3. **Max:** $\tilde{p}((X_l + X_m)/2) = \max(\hat{p}(X_l), \hat{p}(X_m))$

4. **Min:** $\tilde{p}((X_l + X_m)/2) = \min(\hat{p}(X_l), \hat{p}(X_m))$

We find that all 4 have very similar performance to the power-weighted graph, whereas the equivalent estimators using the ground truth density quickly converge. See Figure 10.

## C.3. Trade-off Between Log Density and Score Estimation

We speculate that there is a fundamental trade-off between accurately estimating log densities and their gradients (score functions) that applies to all density estimation methods, including the normalizing flows used in our main experiments. To

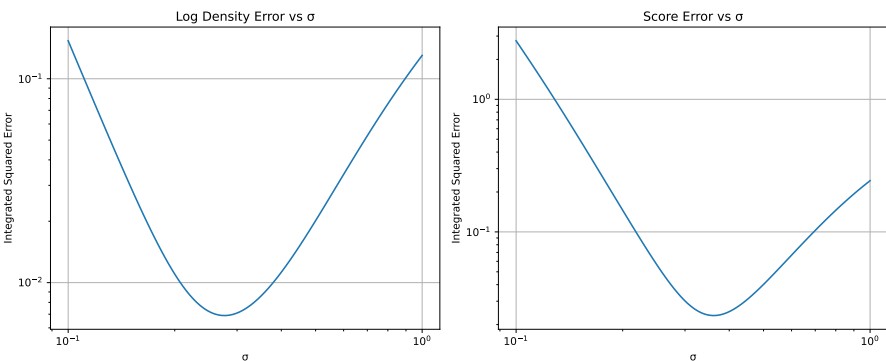

*Figure 11.* Mean integrated squared error (MISE) for log density and score estimation using KDE with varying bandwidth. The optimal bandwidth for log density estimation is smaller than the optimal bandwidth for score estimation.

illustrate this principle in a simple setting where we can compute exact errors, we analyze kernel density estimation (KDE) of the standard normal distribution in one dimension using Gaussian kernels, with 1000 samples drawn from the distribution. While this example is much simpler than the high-dimensional problems and neural network models considered in the main text, it clearly demonstrates how optimizing for density estimation can lead to poor gradient estimates and vice versa.

For a set of samples $\{x_i\}_{i=1}^n$, the KDE estimate with bandwidth $\sigma$ is:

$$\hat{p}(x) = \frac{1}{n\sigma\sqrt{2\pi}} \sum_{i=1}^{n} \exp\left(-\frac{(x-x_i)^2}{2\sigma^2}\right) \tag{34}$$

The log density and score estimates are then:

$$\log\hat{p}(x) = -\log(n\sigma\sqrt{2\pi}) + \log\sum_{i=1}^{n} \exp\left(-\frac{(x-x_i)^2}{2\sigma^2}\right) \tag{35}$$

$$\hat{s}(x) = -\frac{\sum_{i=1}^{n}(x-x_i)\exp\left(-\frac{(x-x_i)^2}{2\sigma^2}\right)}{\sigma^2 \sum_{i=1}^{n}\exp\left(-\frac{(x-x_i)^2}{2\sigma^2}\right)} \tag{36}$$

As shown in Figure 11, the optimal bandwidth for minimizing the mean integrated squared error (MISE) of the log density estimate is smaller than the optimal bandwidth for minimizing the MISE of the score estimate. This reflects a fundamental trade-off: accurate density estimation requires capturing fine details of the distribution, while accurate score estimation requires smoothing out noise to obtain reliable gradients.

Figure 12 visualizes this trade-off. When using the bandwidth optimal for log density estimation, the score estimates are quite noisy. Conversely, when using the bandwidth optimal for score estimation, the log density estimates are oversmoothed. This observation helps explain why we obtain better results using dedicated score models rather than differentiating density models.

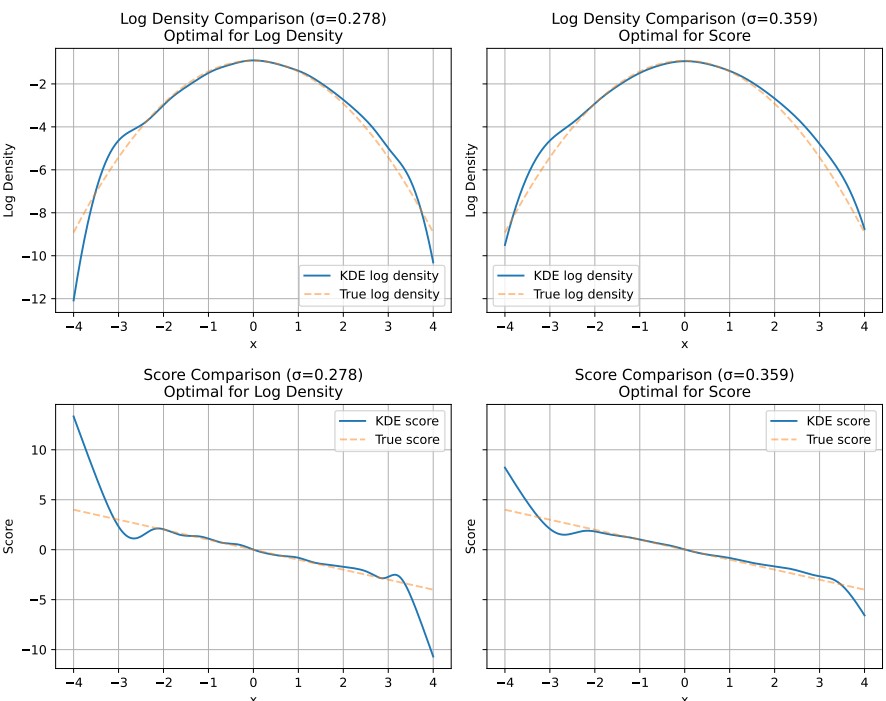

*Figure 12.* Comparison of KDE estimates using different bandwidths. Top row: Log density estimates. Bottom row: Score estimates. Left column: Using the bandwidth that minimizes log density MISE. Right column: Using the bandwidth that minimizes score MISE. Ground truth shown in dotted orange, KDE estimate in solid blue. The optimal bandwidth for log density leads to noisy score estimates, while the optimal bandwidth for score estimation leads to oversmoothed log densities.

