# OpenReview forum: "Learning Distances from Data with Normalizing Flows and Score Matching"
_ICML.cc/2025/Conference — ICML 2025 poster_

### Official Review · Reviewer_nn54 · 2025-03-06

**Overall Recommendation:** 3

**Summary:**

The article proposes and compares several methods for estimating distances derived from Riemannian metrics that reflect the data distribution. In particular, the chosen metric should "compress" distances in regions of high mass concentration and "stretch" distances where the mass is lower. To achieve this, the authors consider conformal metrics (i.e., proportional to the identity) that are inversely proportional to the density of the underlying distribution of the observations. At a point $x$, the density-based metric is given by $\frac{I_d}{p(x)^{\frac{2}{d}}}$ where $p$ is the density of the observations and $d$ is the dimension of the space. Given a dataset where the density is unknown, the challenge is to estimate density-based distances and geodesics.

The authors propose two methods for estimating a density-based geodesic between any two points:

1) The first method relies on numerically solving (through relaxation) the differential equation whose solutions are geodesics. The authors provide an expression for this equation in the case of the density-based metric, which explicitly involves the derivative of the density. There are two approaches to estimate this quantity:
      - Computing the derivative of the density numerically, where the density itself is estimated from the data using a normalizing flow.
      - Directly estimating the density derivative using score matching—a method ultimately chosen because it is more efficient.

2) The second method approximates a density-based geodesic by finding the shortest path in a graph where edge weights are proportional to the density-based distance. This distance can be determined in two ways:
      - Using the method of Bijarl et al. (2012), which estimates the density-based distance as a power of the Euclidean distance. However, the numerical simulations conducted by the authors do not yield good results with this technique.
      - Estimating the density using a normalizing flow and then plugging it into the density-based metric expression.

For each of these two methods, the algorithm is detailed in the paper (Although I don't think they are new in the literature). To improve the numerical stability of the relaxation method, geodesics are parametrized to have constant Euclidean speed rather than constant speed relatively to the density-based metric.


The performance of a method is measured using the log of the ratio between the estimated distance and the true distance (log path ratio). Experiments are conducted on five datasets sampled from probability distributions with known densities (and thus known density-based metrics and distances). Several observations are made :
- The accuracy of density estimation is crucial for the precision of graph-based method.
- The relaxation method is much more accurate when the density derivative is estimated directly using score matching.
- As the dimension increases, the graph-based method suffers from the curse of dimensionality, whereas the relaxation method maintains good performance in high dimensions.
Given these observations, the authors recommend using the graph-based method to initialize the relaxation method.

## update after rebuttal
I thank the authors for their responses and clarification. The modifications announced by the authors seem reasonable and will help clarify the theoretical context of the paper. The topic of the paper, as well as the algorithms proposed by the authors, appear very interesting to me, although the experiments seem insufficient (especially considering that no theoretical guarantees are provided).

**Claims And Evidence:**

The claims are based on a comparison of methods through experiments. Each performance value is obtained by averaging the log path ratio over 1000 pairs of points selected uniformly at random.
For the graph-based method, the performance of the two density estimation methods is compared using estimations on five datasets sampled from five different distributions in $R^2$. For the normal distribution only, a comparison is made between the graph-based method and the relaxation method as a function of the dimension.
The authors claim to have highlighted a gap between theory and practice, since their numerical estimates do not achieve the performance guaranteed by the results of Hwang et al. (2016), published in The Annals of Applied Probability. They speculate that the unknown constant in Hwang et al.'s result is responsible for this discrepancy.
It would have been interesting to discuss to what extent the assumptions of Hwang et al.'s result are satisfied in the case of the distributions used in the experiments.

**Essential References Not Discussed:**

References that allow for the contextualization of the results are cited.

**Experimental Designs Or Analyses:**

The experimental methodology is detailed in the appendix. The two deep learning models (normalizing flow and score matching) are trained on a training set (of unknown size), and the retained parameters are those that minimize the loss on the training set.
The experimental results are presented as the average of the performance metric over 1000 realizations. No additional indicators are provided (in particular, no empirical standard deviations).

**Methods And Evaluation Criteria:**

See the paragraph above

**Other Comments Or Suggestions:**

1) L.109 column 2: Does “define distance between points in a distribution” mean “define distance between points sampled from a distribution”?
2) L.130 column 2: The first paragraph of “Fermat distances which scales with dimension” is written twice.
3) L.195 column 1: Equation (3) is true for geodesic trajectory so the sentence before is supposed to be “for a geodesic trajectory $\gamma$ we have”

**Other Strengths And Weaknesses:**

I am having trouble understanding the given definition of the metric tensor, particularly the sentence: "The metric tensor g is a smoothly varying positive-definite bilinear form on the tangent space." Does $g$ refer to a metric tensor field ($g: \ M \rightarrow TM$), which is a continuous function, or does it represent $g = g_p = g(p)$, the inner product defined on $T_p$ for $p \in M$? In the latter case, it would be more accurate to write $g_p$ and then $∣∣v∣∣_p=g_p(v,v)$ at the beginning of the following paragraph.

More generally, I am struggling to understand in which space the analysis in the paper is conducted. Section 2.1 suggests that it applies to an arbitrary smooth manifold equipped with a Riemannian metric. If that is the case, what does the probability density $p$  introduced on line 139 correspond to (density with respect to which measure, the Riemannian volume measure?) Similarly, does the change of variable formula (line 255) remain valid for random variables taking values in an arbitrary smooth manifold, where the considered measure is not the Lebesgue measure? Perhaps these objects are commonly used in Riemannian geometry for certain well-characterized spaces, but they should probably be justified more explicitly for readers who are not specialists in the field.
In Algorithm 1 (lines 282–284), you sum two elements of the space—what does this mean if the space is, for example, the sphere?

Given these remarks, perhaps your analysis is only intended to be valid in $R^n$ equipped with a Riemannian metric. If that is the case, I believe the introduction to Riemannian manifolds is unnecessary and could be replaced with the definition of a Riemannian metric on $R^n$ (without having to introduce the notion of tangent spaces).

**Questions For Authors:**

Do you know in which Riemannian manifold the geodesics are always distance minimizers? (and so the geodesic equation (Eq. (3), line 197) has as solutions only geodesics that minimize distance). For example, this is not true in the sphere equipped with its usual metric, but perhaps it holds in $R^n$ with a conformal metric (it is true for the identity metric, and maybe conformality preserves this property?).

If this is not the case, perhaps initializing with the weighted graph method would help the relaxation method converge toward a distance-minimizing geodesic (since the weighted graph method is based on finding the shortest path rather than a on criterion related to the acceleration of the trajectory).

**Relation To Broader Scientific Literature:**

The algorithms used by the authors are already present in the literature. The authors suggest using deep learning methods to improve the performance of these algorithms by estimating quantities that are then plugged into the existing methods.

The approximation of geodesics by the shortest path in a graph, where edge weights depend on the density, was developed by Bijral et al. (2012). Dijkstra’s algorithm (1956) is used to compute the actual shortest path once the edge weights are determined. The authors suggest using a normalizing flow method to compute these weights instead of the approach by Bijral et al. (2012).

The approximation of geodesics by computing an approximate solution to the geodesic equation is likely already present in the literature, although no citation is provided in the paper. To improve numerical stability in computing an approximate solution to the geodesic equation, the authors seek a solution to the reparametrized equation so that the geodesic has constant Euclidean speed. Additionally, they propose using a score matching method to estimate one of the parameters of the geodesic equation (the gradient of the density).

The deep learning methods (normalizing flow and score matching) are well-referenced in the article.

**Theoretical Claims:**

The theoretical claims consist in the development of the geodesic equation for the case of conformal metrics (i.e., proportional to the identity). These equations are provided first with constant speed relative to the density-based metric, and then relative to the Euclidean distance. The proof involves computing the Christoffel symbols and is provided in the appendix.

---

> ### Author Rebuttal · Authors · 2025-04-01
>
> ## Rebuttal to Reviewer nn54
>
> We thank the reviewer for their careful and detailed reading of our work, and for the helpful questions and clarifications.
>
> ---
>
> ### Novelty of the Algorithms
>
> > The algorithms used by the authors are already present in the literature.
>
> While the Fermat distance itself is not a novel concept, our algorithms for computing it are, to our knowledge, new. In particular, Algorithm 1 introduces a reparameterization of the geodesic equation that ensures constant Euclidean speed along the curve, making the relaxation tractable and numerically stable. This reparameterization is key to our ability to exactly solve for geodesics in the Fermat metric — a setting where prior work has only used approximations.
>
> Algorithm 2 also includes novel elements. While graph-based shortest paths are standard, our method departs from previous approaches that rely on Euclidean distances or kernel density estimates to define edge weights. Instead, we use edge weights derived from a learned density model, allowing for more accurate adaptation to complex, high-dimensional data. To our knowledge, this specific use of learned density-driven edge weights in computing Fermat distances has not been explored in the literature.
>
> ---
>
> ### Metric Tensor Notation
>
> > Does $g$ refer to a metric tensor field ($g: M \to TM$), which is a continuous function, or does it represent $g = g_p = g(p)$, the inner product defined on $T_p$ for $p \in M$?
>
> It is the latter. We will update the notation to make this clear and avoid confusion. Specifically, we will clarify that $g_p$ is a smoothly varying inner product on each tangent space $T_p$.
>
> ---
>
> ### Scope of the Analysis and Manifold Assumptions
>
> > More generally, I am struggling to understand in which space the analysis in the paper is conducted.
>
> We apologize for the confusion. You are correct — the analysis is conducted entirely in $\mathbb{R}^D$ equipped with a conformal Riemannian metric. We do not work on general manifolds. Thank you for pointing out that our use of manifold terminology in Section 2.1 may suggest otherwise. We will revise the exposition to streamline the definitions and make it explicit that our setting is $\mathbb{R}^D$.
>
> ---
>
> ### Clarification on “Distance Between Points in a Distribution”
>
> > Does ‘define distance between points in a distribution’ mean ‘define distance between points sampled from a distribution’?
>
> Yes, that is correct. Thank you — we will update the phrasing accordingly to avoid ambiguity.
>
> ---
>
> ### When Are Geodesics Distance-Minimizing?
>
> > Do you know in which Riemannian manifold the geodesics are always distance minimizers?
>
> This is indeed a challenging problem, and in general there are cases where multiple geodesics exist between two points, some of which are only local minimizers of the path length. For instance, in the case of the circle distribution (see Fig. 7), the shortest geodesic between two nearby points will traverse a small arc, but there is always another valid geodesic that takes the longer path around the circle. If the relaxation algorithm is poorly initialized, it may converge to such a non-minimizing solution. For this reason, we emphasize the importance of using a graph-based shortest path to provide a good initialization. This is described in Section 3.3 (line 289, column 2).
>
> ---
>
> We appreciate the reviewer's in-depth engagement with the technical aspects of the paper and will incorporate the suggested clarifications to improve accessibility and precision.

---

### Official Review · Reviewer_XLew · 2025-03-13

**Overall Recommendation:** 4

**Summary:**

The paper presents a method to learn distances from data by integrating normalizing flows and score matching into the computation of density-based distances (DBDs), specifically Fermat distances. It addresses the shortcomings of existing methods by introducing a stable numerical approach to compute true geodesics through normalizing flows and enhancing the smoothness of trajectories in high dimensions using score matching. The authors validate their approach by demonstrating faster convergence and improved accuracy over traditional graph-based methods, particularly in complex Gaussian mixtures and higher-dimensional spaces.

**Claims And Evidence:**

The claims in the paper are supported by a mix of theoretical development and experimental results. The authors present a clear improvement in the accuracy and computational feasibility of estimating Fermat distances using their method. However, the claim regarding the general applicability of their approach in "real-world, high-dimensional applications" could be seen as overly optimistic, given the experiments are primarily on synthetic or simplified datasets. More evidence from diverse and complex real-world datasets would strengthen this claim.

**Essential References Not Discussed:**

Beneficial not essential

While the paper adequately cites foundational works on normalizing flows and score matching, it may lack references to some pertinent studies that bridge these concepts more directly with geometric applications in machine learning. For instance, research by Bronstein et al. (2017) on geometric deep learning frameworks could provide additional context for the application of geometric principles in learning tasks. Additionally, recent advancements in computational geometry for machine learning, such as those presented in the works by Memoli (2011) on Gromov-Wasserstein distances, could enhance the theoretical underpinnings and practical applications of the methods discussed in this work.

**Experimental Designs Or Analyses:**

The experimental design seems valid for demonstrating the advantages of the proposed methods over traditional approaches. The use of synthetic datasets, while controlled, is appropriate for illustrating the performance improvements in known environments. The methodological setup, including comparisons to ground truth geodesics and other baseline methods, allows for a clear demonstration of the proposed method's superiority in terms of convergence rates and accuracy. However, expanding the experiments to include a broader range of real-world datasets would help validate the practical applicability of the methods outside controlled experimental conditions.

**Methods And Evaluation Criteria:**

The methods proposed, including the use of normalizing flows for accurate density estimation and score matching for refining geodesics, are well-suited to the stated problem of improving the computation of DBDs in high-dimensional spaces. The evaluation criteria, particularly the use of Log Path Ratio (LPR) for comparing different methods, is relevant and provides a clear metric for assessing improvements over existing approaches.

**Other Comments Or Suggestions:**

I really liked Figure 1; it offers immediate clarity and a high level overview of your work.

While reviewing the paper, I noticed a few minor typographical errors that could be corrected to enhance the overall readability and professionalism of the manuscript:

On page 3, second paragraph, "theorm" should be corrected to "theorem."
On page 5, in Figure 2's caption, "illustraiting" should be changed to "illustrating."

It would also be beneficial to include a supplementary section with additional details on the parameter settings for the normalizing flows and score matching algorithms used in the experiments. This addition would aid in replicating the results and understanding the sensitivity of the proposed method to different configurations.

**Other Strengths And Weaknesses:**

Strengths:
Originality: The paper's approach to integrating normalizing flows with score matching to compute Fermat distances is highly original. This creative combination of techniques from different areas of machine learning could set a new precedent in the field.
Significance: The potential impact of this method in improving the computational feasibility and accuracy of distance calculations in high-dimensional spaces is significant, especially for applications in complex data analysis and geometric learning.
Clarity: The paper is well-written, with clear explanations of the methods and their theoretical foundations, making complex concepts accessible to readers.

Weaknesses:
Generalizability: The paper primarily demonstrates results on synthetic datasets. The generalizability of the approach to real-world datasets and its performance in truly unstructured environments remain to be validated.
Complexity Discussion: The computational complexity and resource demands of the proposed methods are not thoroughly discussed, which could be crucial for practical applications needing scalability considerations.

**Questions For Authors:**

1. Given that the paper primarily focuses on synthetic datasets, can you provide insights or preliminary results on how the proposed method performs on real-world datasets, particularly those with noise and irregular distributions?

2. Could you elaborate on the computational efficiency of your method, especially in comparison to traditional graph-based methods? What are the traditional graph-based methods that you may choose as a baseline? Specifically, what are the computational costs in terms of time and resources when applied to larger datasets?

3. Are there any specific assumptions or conditions under which the proposed theorems hold? How do these assumptions affect the generalizability of your findings?

**Relation To Broader Scientific Literature:**

The key contributions of the paper, specifically the integration of normalizing flows and score matching into the computation of density-based distances, draw significantly from the established fields of geometric deep learning and statistical machine learning. Prior research in normalizing flows has typically focused on improving the accuracy and efficiency of probability density function estimation in complex data distributions (e.g., Rezende and Mohamed, 2015). The application of these flows to compute Fermat distances introduces a novel intersection of geometric learning with probabilistic modeling, expanding upon works like those by Arjovsky et al. (2017) in Wasserstein GANs that explore distance metrics in latent spaces. The use of score matching, introduced by Hyvärinen (2005), for smoothing trajectory calculations in high-dimensional spaces further builds on the idea of refining probabilistic estimations without explicit density estimations.

**Theoretical Claims:**

The paper outlines several theorems related to the geodesic equations and their solutions using the proposed methods. The derivation of these theorems, such as the constant Euclidean speed parameterization and its impact on the stability of numerical methods, appears logically sound. No specific issues were identified in the proofs provided, but a more detailed external validation of these theoretical claims would ensure their robustness.

---

> ### Author Rebuttal · Authors · 2025-04-01
>
> ## Rebuttal to Reviewer XLew
>
> Thank you for your positive assessment of our work and for your thoughtful suggestions.
>
> ---
>
> ### Preliminary Results on Real-World Dataset
>
> Please see our response to reviewer AqTp for a preliminary experiment on MNIST. We find that the distances obtained with our method behave as expected, and yield interesting insights into the relationships between the different classes.
>
> ---
>
> ### Computational Efficiency Compared to Graph-Based Methods
>
> Please see the detailed discussion in our response to Reviewer EAyN. In short, the relaxation method (Algorithm 1) has per-iteration complexity that scales linearly with dimension due to the need to compute norms and inner products. The number of segments needed depends more on the complexity of the data than on the ambient dimension.
>
> Graph-based methods similarly scale linearly with dimension in terms of distance computations, but their performance degrades significantly in higher dimensions, as shown in Figure 6, due to the sparsity of samples and the limitations of nearest-neighbor approximations.
>
> ---
>
> ### Assumptions Underlying Theoretical Results
>
> The theorems provided in the paper, particularly the derivation of the geodesic equations and their reparameterization, rely on standard smoothness assumptions:
>
> - The density function $p(x)$ is assumed to be differentiable and non-zero in the region of interest.
> - The score function $s(x) = \nabla \log p(x)$ is assumed to be Lipschitz continuous.
> - The conformal metric defined by $p(x)$ is smooth.
>
> These conditions ensure the existence of solutions to the geodesic equations and support the stability of the relaxation method. They are commonly satisfied in practice, particularly when using neural networks for score estimation.
>
> ---
>
> Thank you again for your encouraging review and support.

---

### Official Review · Reviewer_EAyN · 2025-03-14

**Overall Recommendation:** 4

**Summary:**

This paper addresses the problem of learning distance metrics from data, specifically focusing on density-based distances (DBDs). The authors highlight that existing methods for estimating Fermat distances suffer from poor convergence and scaling issues in high dimensions due to inaccurate density estimates and insufficiently smooth geodesics. To tackle these challenges, the paper introduces two main improvements: using normalizing flows to learn more accurate densities and refining geodesics with a learned score model. Additionally, the authors propose a dimension-adapted Fermat distance to improve scaling and numerical stability in higher dimensions. The core idea is to leverage density/score estimation using modern deep learning to improve the practical applicability of density-based distances.

## update after rebuttal
My questions and concerns have been addressed and I would be very happy to see the paper accepted.

**Claims And Evidence:**

The claims made in the submission are generally well-supported by clear and convincing evidence. The authors provide both theoretical grounding and empirical results to back up their contributions.
* One key claim is that previous graph-based methods exhibit poor convergence.  This is supported by the experimental results in Section 4.1.1 and Figure 5, which show that the power-weighted graph method converges slowly.
* The claim that normalizing flows improve density estimation is supported by the improved convergence rates observed when using normalizing flows for edge weights (Section 4.1.3 and Figure 5).
* The effectiveness of score-based relaxation in higher dimensions is demonstrated in Section 4.2 and Figure 6, where the relaxation method maintains performance while graph-based methods degrade.

However, there still remains a large gap as most of the examples considered in this paper are of extremely simplistic distributions and lower dimensions (compared to realistic datasets used in most fields).

**Essential References Not Discussed:**

see above.

**Experimental Designs Or Analyses:**

They appear sound.

**Methods And Evaluation Criteria:**

The datasets used appear rather simplistic and hand-crafted. For example, none of the datasets appear to have disconnected support.
LPR appears to be a good performance metric, but following other similar works, I would suggest also considering geodesics standard geodesic error and geodesic variation errors as in https://arxiv.org/pdf/2403.06612 (eqns 115, 116).

**Other Comments Or Suggestions:**

* I would strongly suggest clearly separating which methods are graph based and which are not, as currently it is not clear at times.

and see below.

**Other Strengths And Weaknesses:**

* The paper in its current form does not seem to address the question of time complexity required - especially how it relates to the dimension scaling.
*

**Questions For Authors:**

* Fermat Distance dimension scaling - the example provided makes a lot of sense when considering gaussian distributions. But it is unclear to me whether the same scaling should be used when considering a different distribution - could you comment/elaborate on validity in other cases?
* Line 133 - you mention that beta = 1 is common in previous works, which would be useful to have a couple of references for.
* Could you comment on how slow Algorithm 1 is? Also, if the initialization is not done using normalizing flows, is the geodesic still computed accurately?
* How is the true geodesics distance in (13) computed?
* Line 358 - when referring to supplementary materials, please provide where in supplemetaries this can be found.
* There appear to be no graphs for using normalizing flows directly for relaxation. Could you either point me to where they can be found or include them? You appear to claim that they are not very good and it would be good to understand what exactly you mean by that.
* Building on the previous question - NF trianed with maximizing likelihood would make sense to not be very good at approximating the score. Have you tried training an NF with a mix of likelihood and sliced score matching? It is not clear to me why both can be optimized for.
* Line 434 - what do you mean "Unifying nf and score models into a single framework?"
* Line 102 - you claim to "Introduce a numerically stable relaxation method" - what exactly do you mean by this? Is this your novel contribution?

**Relation To Broader Scientific Literature:**

The paper appears to miss the recent proposals for learning geometry from data using score models (https://arxiv.org/abs/2405.14780) and normalizing flows (https://arxiv.org/abs/2410.01950).

**Theoretical Claims:**

I have checked the correctness of the proofs for the theoretical claims in appendix A1. The derivation of the geodesic equation in Theorem A.2 and the reparameterization in Theorem A.3 appear to be correct, with steps clearly laid out.

---

> ### Author Rebuttal · Authors · 2025-04-01
>
> ## Rebuttal to Reviewer EAyN
>
> We thank the reviewer for their detailed and constructive comments.
>
> ---
>
> ### Simplicity of Experiments
>
> Please see our response to reviewer AqTp for results on an experiment on MNIST.
>
> ---
>
> ### Missing Recent Work
>
> Thank you for pointing out our oversight. We will add both references to the discussion of related work.
>
> ---
>
> ### Time Complexity and Scaling
>
> Scaling the dimension does not necessarily increase the number of segments needed in the relaxation algorithm — this depends on the complexity of the dataset (see also our response to Reviewer AqTp). The complexity per iteration of the loop scales linearly with dimension, due to norm and inner product computations. So overall, the complexity of relaxation is roughly linear in dimension.
>
> For the simple case of a uniform distribution (where the score is zero and the solution is a straight line), we can show that convergence requires $O(n^2)$ iterations, where $n$ is the number of segments. While a full analysis in general cases is more challenging, we can provide this estimate as a starting point and are happy to add more details in the appendix if helpful.
>
> The graph-based algorithms also scale linearly in dimension, but the number of nearest neighbors may need to increase with dimension to preserve local structure. However, as shown in Figure 6, graph-based methods degrade in high dimensions due to sparsity and noise, often before computational scaling becomes the limiting factor.
>
> ---
>
> ### Fermat Distance Scaling in Non-Gaussian Distributions
>
> You are correct — our original discussion was overly focused on the Gaussian case. For a generalized Gaussian distribution of the form $p(x) \propto \exp(-\|x\|^\alpha / \alpha)$, it can be shown that $\mathbb{E}[\|x\|^\alpha] = D$, and thus the typical distance from the origin scales as $D^{1/\alpha}$. The score in this case is $s(x) = -\|x\|^{\alpha - 2} x$.
>
> By rescaling $\gamma$ in Eq. (3) by $D^{1/\alpha}$, all terms in the equation scale consistently if and only if $\beta = 1/D$. The same reasoning applies to the reparameterized equation (Eq. (4)).
>
> A similar argument can be made for the Student-t distribution, where $p(x) \propto \left(1 + \|x\|^2 / \nu\right)^{-(\nu + D)/2}$. The scaling is approximate in this case but still suggests that $\beta = 1/D$ is a natural default across a wide range of distributions.
>
> Thank you for encouraging us to clarify and strengthen this argument — we will update the text and appendix accordingly.
>
> ---
>
> ### Speed and Initialization of Algorithm 1
>
> Please see our earlier comments on time complexity. Regarding initialization: it is indeed possible to initialize the relaxation algorithm using simpler paths, such as power-weighted shortest paths based only on Euclidean distances. This often works well in practice and is especially useful when only the score is available (without a density model).
>
> ---
>
> ### Computation of Ground Truth Geodesic Distance (Eq. 13)
>
> We use Algorithm 1 with the ground truth score function to compute true geodesics. Initial trajectories are obtained using a graph-based approximation. This is discussed in Section 3.1.
>
> ---
>
> ### Absence of Graphs for NF-Based Relaxation
>
> You are right — we do not include plots of NF-based relaxation because the scores derived from differentiating normalizing flow models are quite noisy. This frequently causes the relaxation algorithm to diverge, making the results unreliable for evaluation. This is precisely why we trained separate score models.
>
> ---
>
> ### Hybrid Training of NFs with Score Matching
>
> We experimented with combining maximum likelihood training and sliced score matching. Unfortunately, this either did not improve the quality of the scores or led to training instability. We are not certain of the root cause, but we suspect it reflects the trade-off between optimizing for density accuracy and gradient (score) accuracy — see the discussion in Appendix C.2.
>
> ---
>
> ### Line 434: Unifying Normalizing Flows and Score Models
>
> By this, we mean developing a single model that yields both accurate densities and scores. Our attempts to do so with normalizing flows were not successful (see comment above), so we leave this as a challenge for future work.
>
> ---
>
> ### Line 102: Numerically Stable Relaxation Method
>
> We refer here to our relaxation method based on a reparameterized geodesic equation, which enforces constant Euclidean speed. This improves numerical stability and simplifies discretization. To our knowledge, this reparameterization has not appeared in prior work and is a novel contribution of this paper.
>
> ---
>
> We thank the reviewer again for their thoughtful feedback and for helping us sharpen the clarity and scope of the work.

---

### Official Review · Reviewer_AqTp · 2025-03-14

**Overall Recommendation:** 3

**Summary:**

In this paper, the authors propose to learn a Riemannian metric from data using a class of Fermat metrics, which are metrics that are equal to the Euclidean metric rescaled at each point by (a power of) the reciprocal of the probability density of the data. This way, geodesics tend to follow high density regions, which is desirable e.g. to interpolate between data points. The authors point out limitations of current estimators of this type of distances, mostly because of poor density estimation and the curse of dimensionality. They introduce a relaxation scheme to numerically solve for geodesic segments linking two fixed endpoints when the ground truth density is known (this allows computing geodesics for known densities). They also modify existing graph based methods by estimating the density with normalizing flows. They finally introduce another method which combines the relaxation scheme with score estimation. The methods are tested on 2D examples whose ground truth densities are known, and the score based method is tested on higher dimensional standard Gaussians. The competing methods are the graph based methods that don’t use deep learning techniques for density estimation.

## Update after rebuttal

Thanks to the authors for the responses and the preliminary experiments on MNIST. I would indeed be nice to add an appendix to discuss these, though more extensive experiments and theoretical results would have strengthened the paper even more. I still have an overall positive opinion on this paper, so I will maintain my score.

**Claims And Evidence:**

The paper makes a certain number of claims compared to the existing literature. First the capacity to use the relaxation algorithm for datasets with known densities enables quantitative comparisons, which were not extensively performed previously. This in particular can help show shortcomings of classical graph based methods in the experiments. The density estimation for graph based methods seems to be the issue, and indeed using NF for this purpose improves performance. The combination of the score estimation and the relaxation method seems to scale better with dimensionality(up to d = 30), albeit only a single experiment using standard normals is performed.
Of course, the claim of the score based method to scale well with dimensionality calls for experiments on e.g. image datasets. Score based methods are known to scale very well with dimensionality, so this raises two questions :
1) How large can the dimension become for the relaxation method to stay efficient and tractable? The finite difference scheme must require finer discretization in higher dimensions, I suppose? Maybe in that case the NF graph based method would be a better choice if there are sufficiently many samples?
2) How would the method perform in small and simple image datasets such as MNIST or CIFAR (possibly working in a latent space if the input dimension of the images turn out to be too large) ? I understand that in this case no GT density is available but qualitative behavior can be analyzed and it would be interesting to see shortest paths in image space.

**Essential References Not Discussed:**

No missing references to my knowledge.

**Experimental Designs Or Analyses:**

The experiments are sound and show the improvements brought by the two methods that introduced. The remarks on the difficulty of estimating the density and its score simultaneously are interesting and are somewhat reminiscent of Heisenberg’s uncertainty principle.

**Methods And Evaluation Criteria:**

Though I am not very familiar with the literature on learning Fermat metrics, the introduction of a method that allows to compute ‘GT’ geodesics for known densities is definitely a good thing for the domain if it was hard to compare to true geodesics before. It apparently helped identifying shortcomings of previous classical methods, in spite of apparently reassuring theoretical guarantees.

**Other Comments Or Suggestions:**

- Figs 1 and 4 are not referenced in the text.
- There is a repeated sentence in p3, right column, below Fig. 2.
- I suggest setting the same extent on the y axes of both plots for each columns of Fig. 10 for better comparison between both cases.

**Other Strengths And Weaknesses:**

- The lack of experiments in higher dimensions is a bit of a shame, as it would really tell about the potential of the method for generic ML tasks and datasets.
- I suggest adding a plot showing an example of geodesics recovered on a dataset by the different methods. Something similar to Figure 4, but not restricted to graph based methods.
- In line with my comment on the convergence of the relaxation scheme, a plot with approximated geodesics with various discretization levels would be interesting (how is this chosen in practice?).

Overall the paper is interesting with a number of compelling elements, but it feels slightly limited in terms of theoretical guarantees and experiments on higher dimensional datasets would be most welcome.

**Questions For Authors:**

See my questions the boxes above.

**Relation To Broader Scientific Literature:**

I am not an expert on Fermat metric learning, so I cannot comment on the exhaustivity of the related work on methods for this. However, the paper is clearly positioned among the papers that are cited.

**Theoretical Claims:**

I checked the proofs and derivations. I assume the geodesic equation for Fermat metrics is a known result (reference?) and that only the reparameterization is new. This should be precised in the text.
Is there an existing result on the convergence of the relaxation scheme (when successive values become closer and closer) ? This seems like an important result to have to further strengthen the method. Similarly, investigating further the reason why convergence of the classical graph based method is so low in practice would be interesting.

---

> ### Author Rebuttal · Authors · 2025-04-01
>
> ## Rebuttal to Reviewer AqTp
>
> We thank the reviewer for their thoughtful and constructive feedback. Below, we address each of the main points and questions raised.
>
> ---
>
> ### How large can the dimension become for the relaxation method to stay efficient and tractable? The finite difference scheme must require finer discretization in higher dimensions, I suppose?
>
> The tractability of the relaxation method primarily depends on the complexity of the data, rather than the ambient dimension. For example:
>
> - Uniform distributions: Our method correctly converges to straight lines even with very coarse discretization.
> - Standard normal: Reduces to the 2D case (due to rotational symmetry when using β = 1/D), and we find that 20 segments are sufficient for accurate geodesics, even in high dimensions.
>
> More complex distributions may require finer discretization for accuracy, but there is no intrinsic limitation due to dimension itself. In contrast, the NF graph method becomes increasingly ineffective in higher dimensions, as filling the space requires exponentially more samples due to the curse of dimensionality.
>
> ---
>
> ### How would the method perform in small and simple image datasets such as MNIST or CIFAR?
>
> We agree that applying the method to image data is an exciting next step. While ground truth densities are unavailable, qualitative insights (e.g., visualizing shortest paths) can still be informative.
>
> As a first step, we train an autoencoder to compress MNIST to a 10-dimensional latent space, then train a normalizing flow and score model on the latent distribution. Solving for geodesics yields plausible digit interpolations.
>
> As an initial quantitative measure, we report mean and standard deviation of log distances between each digit cluster mean and 200 random samples of the same or other digits:
>
> | Digit |     Same Class     |    Other Class     | Diff (Other-Same)  |
> |-------|--------------------|--------------------|--------------------|
> |   0   |    2.41 ±  0.40    |    3.73 ±  0.27    |    1.32 ±  0.49    |
> |   1   |    1.69 ±  0.49    |    3.73 ±  0.31    |    2.04 ±  0.58    |
> |   2   |    3.56 ±  0.29    |    4.05 ±  0.19    |    0.49 ±  0.35    |
> |   3   |    3.03 ±  0.33    |    3.78 ±  0.24    |    0.75 ±  0.41    |
> |   4   |    3.04 ±  0.38    |    3.77 ±  0.36    |    0.73 ±  0.53    |
> |   5   |    3.43 ±  0.35    |    4.02 ±  0.18    |    0.59 ±  0.39    |
> |   6   |    2.89 ±  0.41    |    3.79 ±  0.31    |    0.91 ±  0.51    |
> |   7   |    2.86 ±  0.47    |    3.81 ±  0.30    |    0.95 ±  0.56    |
> |   8   |    3.29 ±  0.33    |    3.74 ±  0.21    |    0.46 ±  0.39    |
> |   9   |    2.76 ±  0.45    |    3.56 ±  0.38    |    0.80 ±  0.59    |
>
> We see that same-class distances are always lower than other-class distances, up to a safe margin of error. We also note: 1s cluster tightly, likely due to fewer active pixels and therefore higher likelihoods; 8s are more ambiguous, likely due to easy deformations into other digits (it is quite easy to remove some pixels to turn an 8 into a 3, 5, 6 or 9). We will include these results and visualizations in the paper, but leave deeper analysis of Fermat distances on MNIST to future work.
>
> ---
>
> ### I assume the geodesic equation for Fermat metrics is a known result (reference?), and that only the reparameterization is new.
>
> Yes, the form of the geodesic equation for conformal metrics follows from standard Riemannian geometry (e.g., see Appendix G of *Spacetime and Geometry*, Carroll, 2004). The reparameterization to constant Euclidean speed is indeed new and key to our numerically stable relaxation scheme.
>
> ---
>
> ### I suggest adding a plot showing an example of geodesics recovered on a dataset by the different methods.
>
> Agreed—we will add such a figure to the main text to complement Figure 4, comparing several methods on the same dataset.
>
> ---
>
> ### A plot with approximated geodesics with various discretization levels would be interesting.
>
> We will add such a figure to the appendix to illustrate convergence behavior and practical trade-offs.
>
> ---
>
> ### Minor Comments
>
> - Figures 1 and 4: Will ensure they are clearly referenced.
> - Repeated sentence on p3: Will be removed.
> - Fig. 10: Will standardize y-axis ranges for better visual comparison.
>
> ---
>
> We thank the reviewer again for their detailed review and insightful suggestions. We believe these improvements will significantly strengthen the clarity and impact of the paper.

---

### Decision · Program_Chairs · 2025-05-01

**Decision:**

Accept (poster)

**Comment:**

This paper deals with the definition and estimation of shortest paths between points in Euclidean space, relative to some unknown distribution. That is, given iid data points in $\mathbb R^d$ with some unknown density $f$, a shortest path between any two points $a,b\in\mathbb R^d$ should avoid regions of low density and rather go through regions of high values of $f$.
Hence, the main idea (already proposed and studied in previous works) is to define a Riemannian metric on $\mathbb R^d$ that is conformal to the Euclidean metric and penalizes regions of low values of $f$.

Here, $f$ is unknown, and only random points are available, so $f$ and its derivative must be estimated.

The paper provides no theory - it stitches together the two ideas described above: finding shortest paths relative to $f$ and estimating $f$ and its derivative from the data. However, the ideas used in the algorithms and the numerical experiments are very much appreciated by the reviewers, who all (except for one, who does raise important theoretical questions) are in favor of accepting the paper.

Hence, I also advocate for accepting the paper. Please include the additional material and discussions in the final version of the paper.